

# Wet-dry cycles impact DOM retention in subsurface soils

Olshansky Yaniv, Robert A. Root, Jon Chorover

Department of Soil, Water and Environmental Science, University of Arizona, Tucson, 85721, USA

*Correspondence to*: (Yaniv Olshansky yanivo@email.arizona.edu)

**Abstract.** Transport and reactivity of carbon in the critical zone are highly controlled by reactions of dissolved organic matter (DOM) with subsurface soils, including adsorption, transformation and exchange. These reactions are dependent on frequent wet-dry cycles common to the unsaturated zone, particularly in semi-arid regions. To test for an effect of wet-dry cycles on DOM interaction and stabilization in subsoils, samples were collected from subsurface (Bw) horizons of an Entisol and an Alfisol from the Catalina-Jemez Critical Zone Observatory and sequentially reacted (four batch steps) with DOM extracted from the corresponding soil litter layers. Between each reaction step, soils either were allowed to air dry ("wet-dry" treatment) before introduction of the following DOM solution or were maintained under constant wetness ("continually-wet" treatment). Microbial degradation was the dominant mechanism of DOM loss from solution for the Entisol subsoil, which had higher initial organic C content, whereas sorptive retention predominated in the lower C Alfisol subsoil. For a given soil, bulk dissolved organic C losses from solution were similar across treatments. However, a combination of Fourier transform infrared (FTIR) and near edge X-ray absorption fine structure (NEXAFS) spectroscopic analyses revealed that wet-dry treatments enhanced the interactions between carboxyl functional groups and soil particle surfaces and scanning transmission X-ray microscopy (STXM) data suggested that cation bridging by $Ca^{2+}$ was the primary mechanism for carboxyl association with soil surfaces. STXM data also showed that spatial fractionation of adsorbed OM on soil organo-mineral surfaces was diminished relative to what might be inferred from previously published observations pertaining to DOM fractionation on reaction with specimen mineral phases. This study provides direct evidence of the role of wet-dry cycles in affecting sorption reactions of DOM to a complex soil matrix. In the soil environment, where wet-dry cycles occur at different frequencies from site to site and along the soil profile, different interactions between DOM and soil surfaces are expected and needs to be consider for the overall assessment of carbon dynamics.



## 1 Introduction

Dissolved organic matter (DOM) is the main vehicle of organic carbon and nutrient transport to the subsoil (Kaiser and Kalbitz, 2012; Kalbitz et al., 2000). There it stimulates key biogeochemical processes including heterotrophic microbial activity (Fontaine et al., 2007), mineral transformation, and organic and inorganic nutrient and contaminant mobilization (Chorover et al., 2007; Polubesova and Chefetz, 2014; Zhao et al., 2011). Interactions with subsoil surfaces act to stabilize DOM against advective transport and microbial degradation (Eusterhues et al., 2014; Kalbitz et al., 2000; Lutzow et al., 2006). Furthermore, prior studies have shown that DOM generated in the surface litter layers can be transported preferentially to clay-enriched subsoils via macropore flow paths that bypass the intervening matrix (Rumpel and Kögel-Knabner, 2010). Particularly in semi-arid vadose zones, these DOM-subsoil interactions occur in a context of frequent wet-dry cycles. Although such cyclic conditions likely impact C dynamics, the nature of their effects on micro- to molecular-scale organo-mineral associations remains poorly known.

The principal chemical mechanisms affecting DOM retention at soil particle surfaces – including ligand exchange with surface hydroxyl groups, ion-exchange of organic moieties at charged sites, cation bridging, hydrogen bonding and Van der Waals interactions depend on both DOM molecular composition and mineral surface chemistry (Chorover and Amistadi, 2001; Gu et al., 1994; Kleber et al., 2007, 2014). Interactions of DOM with dissolved polyvalent cations (e.g. $Fe^{3+}$ and $Al^{3+}$) may also result in its coagulation and co-precipitation with nucleating metal (oxy)hydroxides (Chen et al., 2014a; Eusterhues et al., 2011). Drying of OM-mineral complexes can affect the mode of interaction. These effects may include changing of adsorption mode and product surface chemistry. For example, drying can convert OM adsorbate from outer- to inner-sphere coordination (Kang et al., 2008), promote exposure of hydrophobic functional groups of the adsorbed species, and increased surface catalysed transformation reactions (Olshansky et al., 2014). For systems where cation bridging plays a prominent role in DOM adsorption (e.g., to the siloxane surfaces of 2:1 layer type clay minerals), cation charge and valence effects are important, with increasing exchangeable $Ca^{2+}$ relative to $Na^+$ resulting in greater DOM retention (Setia et al., 2013).

Due to the heterogeneous nature of both DOM and soil mineral constituents, fractionation of DOM occurs as a result of a gradient of interaction affinities between the DOM components and various soil particle surfaces (Kaiser et al., 1997; Oren and Chefetz, 2012a). DOM fractionation has been studied extensively on single mineral phases (Chorover and Amistadi, 2001; Vazquez-Ortega et al., 2014) and on bulk soils (Guo and Chorover, 2003; Kaiser et al., 1997; Oren and Chefetz, 2012b). Metal (oxy)hydroxides have been suggested as a dominant adsorbent



for DOM with the result being preferential retention of high molar mass aromatic and carboxylated moieties
(Chorover and Amistadi, 2001; Vazquez-Ortega et al., 2014). Conversely, layered silicates (e.g., smectites,
kaolinite) were reported to adsorb mainly low molar mass and aliphatic DOM fractions (Chorover and Amistadi,
2001; Polubesova et al., 2008). While the use of specimen mineral phases in adsorption experiments facilitates
elucidation of molecular mechanisms of DOM interaction, it does not account for the complexity of competitive
interactions associated with heterogeneous assemblies of weathered surfaces as found in natural soils. Conversely,
using whole soils in adsorption experiments has traditionally hindered mechanistic interpretations of DOM uptake
results. However, increased spatial resolution of spectroscopic methods has helped to overcome these shortcomings
by providing micro- and nano-scale information on both soil-mineral phases and associated organic molecules
(Chen et al., 2014b).

66        The current study aimed to utilize such methodological advances to elucidate: (i) how wet-dry cycles affect

the reactions between DOM and subsoil particle surfaces, and (ii) whether spatial fractionation of DOM is
detectable with nanoscale resolution spectroscopic methods. We hypothesized that discontinuous wet-dry cycling
during DOM reaction with subsoils would increase complexation of carboxyl groups with metal (oxy)hydroxide
surfaces or hydroxylated edge surfaces of aluminosilicate clays and promote association of hydrophobic fractions
with pre-adsorbed and desiccated DOM components relative to a continuous-wet condition. Such wetting-drying
episodes have been hypothesized to affect organic carbon dynamics in water-limited portions of the critical zone,
such as those that occur in the semi-arid southwestern US (Miller et al., 2005; Perdrial et al., 2014), but they have
not been previously investigated in controlled laboratory experiments.
**2 Materials and Methods**
**2.1 Soil samples**
Soils were sampled from below mixed conifer forest in the Santa Catalina Mountains (SCM) and Jemez River
Basin (JRB) Critical Zone Observatories (CZO) in Arizona and New Mexico, respectively (Chorover et al., 2011).
The JRB soil was collected from the south slope of the San Antonio Mountain (35°55'10"N, 106°36'52"W) at an
elevation of 2750 m. The SCM soil was collected from the northeast slope of the zero order basin located in the
Marshall Gulch experimental site (32°25'44"N, 110°46'14"W) at elevation of 2600 m. The mean annual
temperature is 6 and 10.4 °C for the JRB and SCM sites respectively. Both sites are subjected to bimodal annual
precipitation patterns with averages of 850 and 940 mm $y^{-1}$. Parent rock is igneous felsic at both sites; granitic in
the SCM and rhyolitic in the JRB. Therefore, the soils used in experiments developed under similar vegetation and



climatic condition but in different parent materials. The SCM and JRB soils are classified as Typic Ustorthents and
Mixed Psammentic Cryoboralfs, respectively (Soil Survey Staff, 2010 , USDA-NRCS., 1999). Soils were collected
from the litter layer (0-2 cm) and Bw3 horizon (80-100 cm), from pedons excavated in April 2012 and October
2015 for SCM and JRB respectively. The SCM litter layer was collected in October 2015. Soils were air dried and
sieved to obtain the fine earth (< 2 mm) fraction and stored in a closed container. Table 1 presents the bulk
properties of the studied subsoils as measured using standard methods (Sparks, 1996). The mineral assemblages of
both soils were dominated by quartz, feldspars and aluminosilicate clays (Table S1). The SCM soil had higher OM
content ($1.1 \pm 0.5$ mg C mg$^{-1}$) and lower pH ($6.1 \pm 0.04$) than the JRB soil ($0.17 \pm 0.2$ mg C mg$^{-1}$ and $7.05 \pm 0.11$).
**2.2 Dissolved organic matter extraction**
The extraction of DOM was achieved by mixing the air-dried and sieved JRB or SCM litter with ultrapure water
(1:5 g/g), and placing the suspension on a reciprocal shaker at 150 rpm for 24 h. Suspensions were centrifuged at
15,000 $g$ for 30 min to separate the solids, using polypropylene copolymer (PPCO) centrifuge bottles. Adsorption
or contamination of DOM from these bottles was measured to be negligible (Vazquez-Ortega et al., 2014). The
supernatant solution was transferred into 50 mL PPCO centrifuge tubes and centrifuged again at 40,000 $g$ for 20
min to remove colloidal organic material and the inorganic clay fraction. Supernatant solutions were filtered
through pre-combusted and cleaned 0.7 µm glass fiber filters. TOC was measured immediately after extraction
(Shimadzu TOC-VCSH, Columbia, MD) and solutions were diluted using ultrapure water to give initial DOC
concentrations of 45 mg L$^{-1}$ (Table 1). DOM solutions were stored at 4°C prior to use.
**2.3 Sequential batch experiments**
To model the effect of sequential hydrologic events delivering litter leachate to subsoils in the two CZO sites,
subsoils were reacted in a set of four steps with DOM extracted from the litter layer of the corresponding profile.
Thirty mL aliquots of DOM ([DOC] = 45 mg L$^{-1}$) solution were mixed with 3.0 g of soil in 50 mL PPCO centrifuge
tubes and agitated (150 rpm, orbital shaker) at room temperature, in the dark. Preliminary kinetic experiments
indicated an apparent equilibration time of 98 h, and this was chosen as the equilibration time for each reactor
vessel. Suspensions were centrifuged for 30 min at 40,000 g and 28 mL were filtered through precombusted 0.7
µm glass fiber filters and the solutions were stored at 4 °C for a maximum of 24 h prior to analysis, as discussed
below. For *continually-wet* treatments, a fresh 28 mL aliquot of DOM solution was added to each tube and
suspensions agitated for an additional 98 h (28 mL were used because ca. 2 mL remained as entrained solution in
the wet soil paste). For *wet-dry* treatments, the soil pastes were air dried for 24 h (drying was accomplished by



directing a low-flow circulating dry-air stream to promote desiccation), then an aliquot of 30 mL DOM solution
was added to each tube and suspensions were re-agitated for 98 h, for a total of four sequential reaction cycles.
Three replicates were prepared for each soil and treatment combination. After the four sequential reaction cycles,
soils were freeze-dried and TOC and TN were measured using ECS 4010 CHNSO Analyzer (Costech, MI, Italy).
During the experiment samples were maintained under oxic condition by equilibration with oxygenated headspace.
**2.4 Characterization of DOM solutions before and after reaction**
Reacted and unreacted DOM solutions were characterized by the following suite of complementary analytical
methods: soluble TOC and TN were determined by total elemental analyzer (Shimadzu TOC-L and TNM-L,
Columbia, MD), absorbance spectra (190 to 655 nm) were collected using a UV-Vis spectrometer (Shimadzu
Scientific Instruments UV-2501PC, Columbia, MD, USA), fluorescence excitation–emission matrices (EEM) were
obtained with a FluoroMax-4 equipped with a 150 W Xe-arc lamp source (Horiba Jobin Yvon, Irvine CA, USA),
and Fourier transform infrared (FTIR) spectra were collected using a Nicolet NEXUS 670 IR spectrometer
(Madison, WI). The EEMs were acquired with excitation (Ex) from 200 to 450 nm and emission (Em) from 250
to 650 nm in 5 nm increments. Spectra were collected with Ex and Em slits at 5- and 2-nm band widths,
respectively, and an integration time of 100 ms. Ultrapure water blank EEMs were subtracted and fluorescence
intensities were normalized to the area under the water Raman peak, collected at excitation 350 nm. Additionally,
an inner-filter correction was performed based on the corresponding UV–Vis scans (Murphy et al., 2013).
Transmission FTIR spectra were collected with a KBr beam splitter and a deuterated triglycine sulfate (DTGS)
detector. Aliquots of two mL of JRB DOM solutions were transferred onto IR transmissive Ge windows, dried
under vacuum for 19 h, and spectra collected in transmission mode. For SCM DOM, 2 mL aliquots were freeze
dried and mixed with IR-grade KBr, then compressed into pellets. For each sample, 120 scans were collected over
the spectral range of 400–4000 cm$^{-1}$ at a resolution of 4 cm$^{-1}$. Clean Ge windows and KBr pellets were used as
background.
**2.5 Scanning Transmission X-ray Microscopy and Near Edge X-ray Adsorption Fine Structure (STXM-**
**NEXAFS) analysis of soils**
STXM-NEXAFS analyses were conducted on clay-size isolates to avoid particulate organic matter and to overcome
possible alteration of C speciation during preparation of thin sections (Chen et al., 2014b). Clay size fractions (<2
μm) of the reacted and unreacted JRB soils were separated by sedimentation after dispersion in ultrapure water
using a sonication bath. Samples for STXM analysis were prepared by depositing 5 μL of diluted aqueous



suspension onto a $Si_3N_4$ window (75 nm thick) and air-dried. The samples were analyzed by STXM on beamline
10ID-1 at the Canadian Light Source (CLS), a 2.9 GeV third-generation synchrotron source. The microscope set
up used a 25 nm Fresnel zone plate, which provided a maximum spatial resolution of *ca*. 30 nm. Samples were
kept under 1/6 atm of He during measurement.
Spatially resolved spectra obtained by collecting stacks of images at energies below and above C 1s, Ca
2p, Fe 2p, element edges. The dwell time was set to 1 ms and pixel sizes of 150 nm. Incident energy was calibrated
with $CO_2$ at 290.74 eV.
The aXis2000 software package (Hitchcock et al., 2012) was used for STXM image and spectral
processing. Stacks were aligned and converted to optical density using a clean area of the $Si_3N_4$ window for
normalization. Regions of interest (ROI) of C, Ca and Fe were extracted from each stack by subtracting below the
edge from the optical density (OD) maps. C NEXAFS spectra were extracted by averaging the pixels from the ROI.
NEXAFS spectra were normalized and peak deconvolutions were performed using the ATHENA software package
(Ravel and Newville, 2005). Peak assignments were based on Cody et al. (1998, 2008), Myneni (2002) and
Urquhart et al. (1997).

**2.6 Data analysis**

Statistical analyses were performed using *R* software packages (Mangiafico, 2016). Data were checked for
normality and equal variance. Means were tested using Kruskal–Wallis for non-parametric analysis. The
differences between means were examined using Dunn test for non-parametric analysis.
The specific UV absorbance ($SUVA_{254}$) was calculated by normalizing absorbance at incident wavelength
254 nm by the cell path length (1 cm) and DOC concentration (M). Fluorescence index (FI, Eq. 1) and humification
index (HIX, Eq. 2) values were calculated from the corrected EEMs (McKnight et al., 2001; Ohno, 2002) as
follows:
$$FI_{Ex370} = \frac{I_{450}}{I_{500}}$$ (1)
$$HIX_{Ex255} = \frac{\sum(I_{435\to480})}{\sum(I_{300\to345})}$$ (2)
where Ex is the excitation wavelength (nm) and *I* is the fluorescence intensity at each wavelength.



Spectra collected by FTIR were background corrected using KBr pellets or the Ge transmission window
as blanks and baseline corrected using the spline function in the OMNIC 8 software program (Thermo Nicolet Co.,
Madison, WI). Peak positions were determined using the second-order Savitzky–Golay method. Voigt line shape,
(a convolution between mixed Gaussian and Lorentzian line shapes) were fitted to the peaks in the 850-1850 cm$^{-1}$
region using Grams/AI 8.0 spectroscopy software (Thermo Electron Corporation). Peak assignments were based
on Socrates (2004), Mayo et al. (2004), Omoike and Chorover et al. (2004) and Abdulla et al. (2010).

## 3. Results

### 3.1 Total Carbon and Nitrogen

The loss of DOC from solution per unit mass of soil was largely independent of reaction step and treatment.  The
mass loss of DOC upon reaction with SCM soil was $156 \pm 5$, $217 \pm 3$, $167 \pm 17$, and $192 \pm 10$ mg kg$^{-1}$ for steps 1-
4, respectively, in the wet-dry treatment, and $163 \pm 3$, $222 \pm 4$, $217 \pm 2.5$, and $214 \pm 6$ mg kg$^{-1}$ in the continuously-
wet treatment. The mass loss of DOC upon reaction with JRB soil was $248 \pm 19$, $257 \pm 1$, $197 \pm 5$, and $200 \pm 12$
mg kg$^{-1}$ for steps 1-4, respectively, in the wet-dry treatment, and $256 \pm 7$, $236 \pm 26$, $176 \pm 44$, and $208 \pm 2$ mg kg$^{-1}$
in the continuously-wet treatment. Hence, the mean fraction of OC removed from DOM solution was $58 \pm 5$ %
(SD) after each reaction step with JRB soil and OC uptake values were not significantly different between the
continuously-wet and wet-dry treatments. In the SCM soil, the mean fraction of OC removed was $41 \pm 4\%$ of the
total after each reaction step in the wet-dry treatment. In contrast to the other three treatments, the continually-wet
SCM treatment indicated increasing amounts of OC removed in each step, with $39 \pm 0.8\%$ in the first step, $48 \pm$
1% in the second, and $56 \pm 1\%$ in the third and fourth steps (Figure 1). At the end of four reaction steps the TOC
of JRB soils increased from $1{,}700 \pm 74$ mg OC kg$^{-1}$ for the unreacted soil to $2{,}750 \pm 87$ mg OC kg$^{-1}$  and $2{,}840 \pm$
99 mg OC kg$^{-1}$ for the wet-dry and continuous-wet treatments respectively (Figure 1). For the JRB soil, increases
in solid phase organic C are in close agreement with the cumulative amounts of DOC removed from reacted
solutions ($902 \pm 26$ and $876 \pm 34$ mg OC kg$^{-1}$ for wet-dry and continuous-wet treatments respectively) and represent
a 60% increase in soil TOC. Conversely, for the SCM soil, despite comparable cumulative losses from solution
($733 \pm 29$ and $817 \pm 2$ mg OC kg$^{-1}$ for wet-dry and continuous-wet treatments respectively), solid phase analyses
indicated that the OC content of the reacted SCM ($11{,}200 \pm 380$ and $11{,}200 \pm 290$ mg OC kg$^{-1}$ soil for wet-dry and
continuous-wet treatments respectively) soils were effectively unchanged relative to the unreacted control ($11{,}800$
$\pm 180$ mg OC kg$^{-1}$).



Patterns in the removal of total N from the DOM solutions showed similar trends for both soils. In the first
two wet-dry steps, a higher proportion of TN was removed from the solution (65 - 70% and 50 - 66% for SCM and
JRB soils, respectively) than in the third and fourth steps (31 - 44% for both soils). The measured increase in soil
TN by the end of the experiment were 63 and 143 mg N kg soil$^{-1}$ for SCM and JRB soils respectively. These values
are slightly higher than the sum of TN removed from the solution (51 and 88 mg N kg soil$^{-1}$ for SCM and JRB soils
respectively) (Figure 1).
The C:N ratio for all reacted DOM solutions decreased from step 1 to step 4, indicating preferential loss of
C from solution, with no significant difference between the continually-wet and wet-dry treatments. However, after
the first reaction with the SCM soil, the C:N ratio was $22.0 \pm 1.3$, which was higher than the unreacted DOM ($14.1$
$\pm 0.8$). It is important to note that DOM extracted from unreacted soil had a C:N ratio of $23.7 \pm 0.9$, and C:N of
DOM decreased during the sequential reaction steps. After the fourth reaction step, ratios of $11.1 \pm 0.8$ and $9.6 \pm$
$0.8$ were observed for the wet-dry and the continually-wet treatments respectively. The C:N of the reacted DOM
solution with JRB soil decreased from $10 \pm 1.0$ after the first reaction step to $4.6 \pm 0.5$ after the fourth reaction step.
The C:N ratio of unreacted DOM solution was $8.4 \pm 0.8$. The overall change in soil C:N ratio was evaluated by the
differences between unreacted soil and soils reacted four times with DOM solutions (Figure 1).  Reacted SCM soils
had significantly lower C:N ($24.2 \pm 1$) than unreacted SCM soil ($30.5 \pm 1.8$). However, no change in C:N was
detected for reacted versus unreacted JRB soils.
**3.2 UV-Vis and Fluorescence Spectroscopy**
Reaction with subsoils altered spectroscopic properties of the litter-derived DOM solutions as reflected in UV-Vis
($SUVA_{254}$) and fluorescence indices (HIX and FI), and there was relatively little variation between continually-wet
and wet-dry treatments (Figure 2). For both JRB and SCM the $SUVA_{254}$ values of DOM decreased (relative to
unreacted DOM) upon contact with soil (Figure 2), with the exception of the fourth step in wet-dry treatment of
SCM soil (Figure 2). This effect of contact with soil on $SUVA_{254}$ was larger for JRB than SCM, although it
decreased with progressive reaction steps even for JRB soils from *ca*. 200 (L mol$^{-1}$ cm$^{-1}$) in the first step to *ca*. 50
(L mol$^{-1}$ cm$^{-1}$) by the fourth step. High $SUVA_{254}$ ($905 \pm 35$ L mol$^{-1}$ cm$^{-1}$) was measured for DOM extracted from
unreacted JRB soil (Table 1).  We note that $SUVA_{254}$ values of unreacted DOM also decreased between the first
(393 L mol$^{-1}$ cm$^{-1}$) and subsequent steps (~350 L mol$^{-1}$ cm$^{-1}$) indicating some alteration of DOM chromophores in
the stock DOM solution during the experiment. Although this was a small change relative to soil reaction effects,
alteration was also evident in the HIX of unreacted JRB DOM. Therefore, treatment effects (continuous-wet and
dry-wet) were evaluated on the basis of differences between reacted and unreacted solutions for the same reaction



step. The effect of reaction with soil on SUVA$_{254}$ values were less pronounced for SCM relative to JRB soils.  In
the wet-dry treatment of SCM soil, SUVA$_{254}$ values of the first three steps were generally consistent at *ca*. 330 ±
13 (L mol$^{-1}$ cm$^{-1}$) and in the fourth step the SUVA$_{254}$ increased to 530 ± 2 (L mol$^{-1}$ cm$^{-1}$).  Conversely, SUVA$_{254}$
values increased slightly over the course of the experiment from 324 ± 10 to 410 ± 16 L mol$^{-1}$ cm$^{-1}$ for the
continually-wet SCM treatment.
Humification index (HIX) values for the reacted DOM were generally higher or similar to the unreacted
DOM (Figure 2). As with the SUVA$_{254}$ index, the fourth step of SCM wet-dry treatment was the exception (Figure
2), giving a lower HIX for reacted compared to unreacted DOM. The HIX values for DOM reacted with JRB soil
were similar for continually-wet and wet-dry treatment. Conversely, with SCM soil, values for the wet-dry
treatments were lower than for continually-wet treatments. The relative differences between reacted and unreacted
DOM were lower for the JRB system than for the SCM system. For both JRB and SCM soils, higher fluorescence
index (FI) values were observed for reacted relative to unreacted DOM (Figure 2) whereas wet-dry versus wet-only
treatment effects were negligible. For JRB, FI values increased from 1.31 ± 0.04 (unreacted DOM) to 1.53 ± 0.04
whereas corresponding values for SCM were 1.34 ± 0.04 and 1.42 ± 0.02, respectively. All FI values are in close
agreement with the value of DOM associated with predominantly plant material (*ca*. 1.4), as opposed to microbial-
derived DOM (*ca*. 1.9) (McKnight et al., 2001).
**3.3 FTIR**
Transmission FTIR spectra of reacted and unreacted DOM for the JRB and SCM systems are shown in Figures 3
and 4, respectively. The most prevalent peaks in the spectra were associated with amide I and II (1636 and 1560
cm$^{-1}$, respectively), carboxylate (asymmetric and symmetric stretches at 1592 and 1417 cm$^{-1}$, respectively), alkyl
(CH$_2$ and CH$_3$ bending vibrations at 1455 and 1380 cm$^{-1}$, respectively), and aromatic moieties (C=C ring vibration
at 1500 cm$^{-1}$, phenol O-H bend 1370 cm$^{-1}$) and O-alkyl (CO$^-$ stretch at 1030-1150 cm$^{-1}$).
For JRB soil, the first reaction step in both continually-wet and wet-dry treatments was accompanied by a
decrease in peak intensities of carboxylate (1592 cm$^{-1}$ and 1417 cm$^{-1}$) and amide (1636 and 1560) relative to O-
alkyl (1150-1030 cm$^{-1}$). Additionally, primary alcohol (1035 cm$^{-1}$) peak intensity decreased relative to secondary
alcohol (1100 cm$^{-1}$). This trend persisted in the second step with JRB soil for both treatments, although the pattern
was less pronounced and differed by treatment. Specifically, the wet-dry treatment showed a larger decrease in the
asymmetric carboxylate stretch (1592 cm$^{-1}$) whereas the continuous-wet treatment showed a larger decrease in the
amide I peak (1636 cm$^{-1}$). In the third step, the decrease in amide and carboxyl peaks relative to O-alkyl was not as
pronounced for the wet-dry as it was in the continually-wet treatment. Finally, in the fourth step of the wet-dry



system, a pronounced decrease in amide and carboxyl peaks relative to O-alkyl was again observed, whereas it was
not in the continually-wet treatment (Figure 3).
Figure 4 shows the spectra of reacted and unreacted DOM in the SCM system. The SCM DOM spectra
show similar peaks as the JRB with the addition of carboxyl (C=O stretch at 1720 cm$^{-1}$) and ester (C=O stretch
1770 cm$^{-1}$ and C-O stretch 1265 cm$^{-1}$). Similar to the JRB system, after reaction with soil, the peaks associated with
carboxyl, carboxylate and amide decreased relative to the O-alkyl peaks and this trend was more pronounced in the
first step than in the subsequent steps. Similar to the JRB system, in the fourth step of the wet-dry treatment, a
pronounced decrease in carboxyl, carboxylate and amide peaks was again observed relative to the O-alkyl peaks.

## 3.4 STXM-NEXAFS

Given limitations in beam time, synchrotron analyses were focused on the JRB soil because it showed larger carbon
accumulation over the course of the experiment. Scanning transmission x-ray microscopy (STXM) images of C,
Fe and Ca obtained for the isolated fine fraction of JRB soils reacted four times with DOM in wet-dry and
continually-wet treatments are shown in Figures 5 and 6, respectively. The carbon signal was observed over all
particle surfaces, from continually-wet and wet-dry treatments after four reaction steps. Locations of higher Fe and
Ca content were observed for both treatments. Near edge x-ray absorption fine structure (NEXAFS) spectra
extracted from C, Ca and Fe-rich regions of interest (ROI) of the STXM maps and C NEXAFS spectra of bulk
unreacted soil and DOM are included in Figures 5 and 6. Spectra of the unreacted DOM consist of peaks
representing aromatic (1s→π* at 285.1 eV), alkyl (1s→3p/σ* at 287.5 eV), amide (1s→π* at 288 eV), carboxyl
(1s→π* at 288.5 and 290 eV), O-alkyl (1s→π* at 289.5 eV) moieties. The C NEXAFS spectra of unreacted soil
show no strong peaks of amide, carboxyl and O-alkyl, similar to the unreacted DOM spectra. However, after four
steps of reaction with DOM, soil from both continually-wet and wet-dry treatments exhibited greatly enhanced
carboxyl and O-alkyl peaks relative to the unreacted soil. In the wet-dry treatment, the aromatic peak was absent.
The O-alkyl peak was more pronounced for the continually-wet than for the wet-dry treatment. Additionally, the
amide peak was suppressed in the reacted soil compared to the unreacted DOM, and for the wet-dry treatment this
peak was absent and was not included in the fitted spectra (supplementary material). The C NEXAFS spectra of
Ca and Fe enriched ROIs are similar to the average whole image spectra. However in the Ca ROI, the carboxyl
peak intensity was enhanced relative to Fe ROI and the averaged whole image spectra. This carboxyl enhancement,
which was absent in the unreacted soil, was most pronounced in the wet-dry treatment.
Variations in the C NEXAFS spectra of the reacted soils following each reaction step are displayed in
Figure 7. After the first reaction step, intensities of the carboxyl and O-alkyl peaks were relatively increased. For



the continually-wet treatment, spectra collected following the second and third steps show an increase in alkyl and
O-alkyl peaks, whereas this trend was less evident in the wet-dry treatment.

### 4. Discussion

Specific surface area (SSA) and OC content are dominant factors controlling sorption of DOM to soil. For
comparable mineralogy, higher SSA tends to increase DOM sorption, while higher solid phase OC content
suppresses it (Kaiser et al., 1997; Oren and Chefetz, 2012b). In addition, solution chemistry can control DOM-soil
interactions. For example, low pH can neutralize weakly acidic OM functionalities, thereby decreasing electrostatic
repulsion from negatively-charged surfaces, whereas bivalent cations such as $Ca^{2+}$ can form bridging complexes
between negatively-charged surface and DOM sites (e.g., Setia et al., 2013). Further, the presence of polyvalent
metal cations in solution can promote precipitation of (meta-)stable OM-metal complexes (Kleber et al., 2014). In
the current study, in spite of differences in soil constituents and DOM compositions deriving from the two distinct
CZO sites, similar amounts of DOM were removed from solution with both JRB and SCM soils. The fact that OC
did not accumulate in the SCM soil solid phase despite significant removal from solution suggests that
decomposition and mineralization are dominant factors indicated in the removal of OC from the reacted SCM DOM
solutions. Indeed, the pronounced decrease in C:N ratio of the reacted soil is consistent with microbial
transformation of organic matter (German et al., 2011). Higher HIX for all SCM reacted samples, with the
exception of the last step in the wet-dry treatment, further support OM transformation. Enhanced mineralization in
the SCM relative to JRB soil may be related to its substantially higher native OC content (Table 1), which would
preclude surface stabilizing interactions and support a significantly higher native heterotrophic microbial biomass.
The relatively lower HIX value for the last step of wet-dry treatment coincide with higher $SUVA_{254}$. Since $SUVA_{254}$
index is correlated with sample aromaticity (Weishaar et al., 2003), an increase in the aromatic peak in the FTIR
spectra was expected. However, FTIR spectra show a relative increase in O-alkyl rather than the aromatic
vibrations. It is possible that the decrease observed in 1550 to 1700 $cm^{-1}$ region is mainly due to a decrease in
carboxyl associated peaks rather than increased aromaticity. It is unclear if the removed fraction was exchanged
with previously adsorbed OM or preferentially decomposed in the solution.

312        Conversely, significant DOM or soil organic matter decomposition was not observed for the JRB soil

experiments, as evidenced from the C mass balance. Therefore, changes in reacted DOM composition can be
attributed to preferential adsorption and exchange reactions. The increased FI value of the reacted DOM further
suggests preferential adsorption of plant- relative to microbial-derived OM. The slight decrease in $SUVA_{254}$ values



is also consistent with this observation, since polyphenols derived from lignin account for most of the aromaticity
in DOM.

318        Spectra from C-NEXAFS obtained for the JRB soil fine fraction corroborate the solution data obtained by

FTIR. A pronounced increase in the carboxyl peak (288.5 eV) after the first reaction step (Figure 7) is consistent
with the decreased intensity of carboxyl in the reacted DOM solutions (Figure 3). NEXAFS spectra collected after
the second and third steps of both treatments show additional increases in the O-alkyl (289.5 eV) and alkyl (287.5
eV) that corroborate the relative decrease in FTIR peak intensities for these functionalities. The fact that the
NEXAFS of the reacted JRB soils clearly shows a relative increase in the carboxyl peak from the third to the fourth
step in the wet-dry treatment (Figure 7), suggests that preferential adsorption of the carboxylic component was
facilitated by the pre-existing soil-DOM phases of the dried soil. Prior work has shown that soil drying may promote
conformational changes in pre-adsorbed DOM that promotes preferential desorption of O-alkyl relative to further
inner-sphere coordination of carboxyl components (Kang et al., 2008; Kang and Xing, 2007). Additional support
for the formation of inner-sphere carboxyl complexes is from the higher preferential adsorption of carboxyl over
amide as observed in FTIR spectra of wet-dry compared to continuous-wet treatments (Figure 3).

330        Due to the heterogeneous composition of soil surfaces and DOM, spatial fractionation of the adsorbed

carbon moieties was expected. Figures 5 and 6 show that in both wet-dry and continuously-wet treatments, regions
containing higher content of Fe and Ca can be distinguished. Interestingly, the carbon NEXAFS spectra of these
distinct locations are generally similar. It is important to note that low Fe spectral signals were detected over all of
the particle surfaces images with STXM. This may suggest that weathered surfaces of the soil, coated with a thin
layer of metal (Fe) oxides and organic matter, can smear out what might otherwise be observed as a spatial
fractionation at this scale (nm).

337        However, close inspection of the C spectra extracted from Fe and Ca enriched zones and whole particle

regions reveal that in samples treated with wet-dry steps, the amplitude of the carboxyl peak shows a relative
increase preferentially in the Ca enriched regions (Figure 5 and supporting information). This finding suggests that
cation bridging interactions are pronounced in stabilizing the carboxyl component in the studied soil. It is important
to note that the solution pH was close to 7, and therefore deprotonated carboxylate species were predominant in
the suspension. Regions of high Ca are likely associated with charged aluminosilicate surfaces hosting
exchangeable cations. The enhancement effect of drying on Ca-carboxylate complex formation can be related to
the tendency of the $Ca^{2+}$ hydration shell to become more acidic upon drying (Sposito, 1984). As water molecules
are gradually removed during air drying, polarizing forces of the $Ca^{2+}$ cation increases, enhancing the tendency of
hydration water to donate protons (Dowding et al., 2005). Therefore, upon drying, protonation of the carboxylate





functionality is expected. Protonation of carboxylate decreases the electrostatic repulsion from negatively charged clay surfaces and increases the overall interaction with clays. It is important to note that our studied soils are predominantly composed of silicate and aluminosilicate minerals and are relatively depleted in crystalline and short range order metal oxides.

**5. Conclusion**

Results of this study show that wet-dry cycles affect interactions between DOM and subsurface soils, in this case by enhancing the interactions between carboxyl functional group and soil surfaces. Interactions of these functionalities were dominated by $Ca^{2+}$ bridging to soil surfaces. The data also demonstrate that nanoscale spatial fractionation of DOM on soil organo-mineral surfaces was diminished relative to what might be inferred from observations pertaining to DOM fractionation on specimen mineral phases. This is likely due to the heterogeneous composition of the weathered soil surfaces and passivation of the underlying mineralogy by metal oxide and OM films. Fractionation of DOM in solution was similar under wet-dry conditions for a soil that presented measureable decomposition of the DOM (SCM) as it was for a soil that did not show any detectable decomposition (JRB).

This study provides direct evidence of the role of wet-dry cycles in the sorption reactions of DOM to a complex soil matrix. In the soil environment, where wet-dry cycles occur at variable frequencies from site to site and along the soil profile, different interactions between DOM and soil surfaces are expected. This wet-dry effect can partially explain the observation that carbohydrates predominate in subsoil horizons, were soil is less subjected to drying, whereas aromatic and carboxylic compounds are more prevalent in top soils, where wet-dry cycles are more frequent (Kaiser and Kalbitz, 2012). Our findings demonstrate the need to consider the effect of wet-dry cycles in studying the interactions between DOM and soil surfaces.

**Acknowledgements:** This research was funded by the Binational Agricultural Research and Development (BARD) program, postdoctoral fellowship to Y. Olshansky grant no. FI-534-2015, and the National Science Foundation, grant no. EAR 13-31408, which supports the Catalina-Jemez Critical Zone Observatory. The STXM analysis described in this paper was performed at the Canadian Light Source beamline 10ID-1, which is supported by the Canadian Foundation for Innovation, Natural Sciences and Engineering Research Council of Canada, the University of Saskatchewan, the Government of Saskatchewan, Western Economic Diversification Canada, the National Research Council Canada, and the Canadian Institutes of Health Research. Thanks to Mary Kay Amistadi, Rachel Nadine Burnett and Prakash Dhakal for assistance with analysis.




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



**Table 1. Physico-chemical characteristics of the study soils**

|  | JRB | SCM |
|---|---|---|
| Clay (%) | 33.9 | 22.6 |
| Silt (%) | 27.7 | 38.4 |
| Sand (%) | 38.4 | 50.9 |
| SSA ($m^2\,g^{-1}$) [a] | $16.6 \pm 0.2$ | $7.7 \pm 0.1$ |
| CEC ($mmol_c\,kg^{-1}$) [b] | $86.6 \pm 4.2$ | $61.3 \pm 0.8$ |
| OC (%) [c] | $0.17 \pm 0.02$ | $1.11 \pm 0.5$ |
| pH [d] | $7.05 \pm 0.11$ | $6.10 \pm 0.04$ |
| EC ($\mu S\,cm^{-1}$) [d] | $61.5 \pm 26.6$ | $36.8 \pm 8.8$ |
| DOC ($mg\,L^{-1}$) [d] | $3.59 \pm 0.82$ | $13.45 \pm 1.30$ |
| DOM pH | $6.97 \pm 0.06$ | $5.91 \pm 0.11$ |
| DOM EC ($\mu S\,cm^{-1}$) | $170.7 \pm 10.2$ | $84.1 \pm 12.3$ |
| SUVA ($L\,mol^{-1}\,cm^{-1}$) | $905 \pm 35$ | $539 \pm 105$ |
| HIX [e] | $1.5 \pm 0.1$ | $4.5 \pm 2.3$ |
| FI [f] | $1.40 \pm 0.04$ | $1.43 \pm 0.03$ |


[a] BET-$N_2$ Specific surface area
[b] Cation exchange capacity
[c] Organic carbon
[d] Obtained in soil aqueous extract (1:10 with 8.2 MΩ, Barnstead water)
[e] Humification index
[f] Fluorescence index



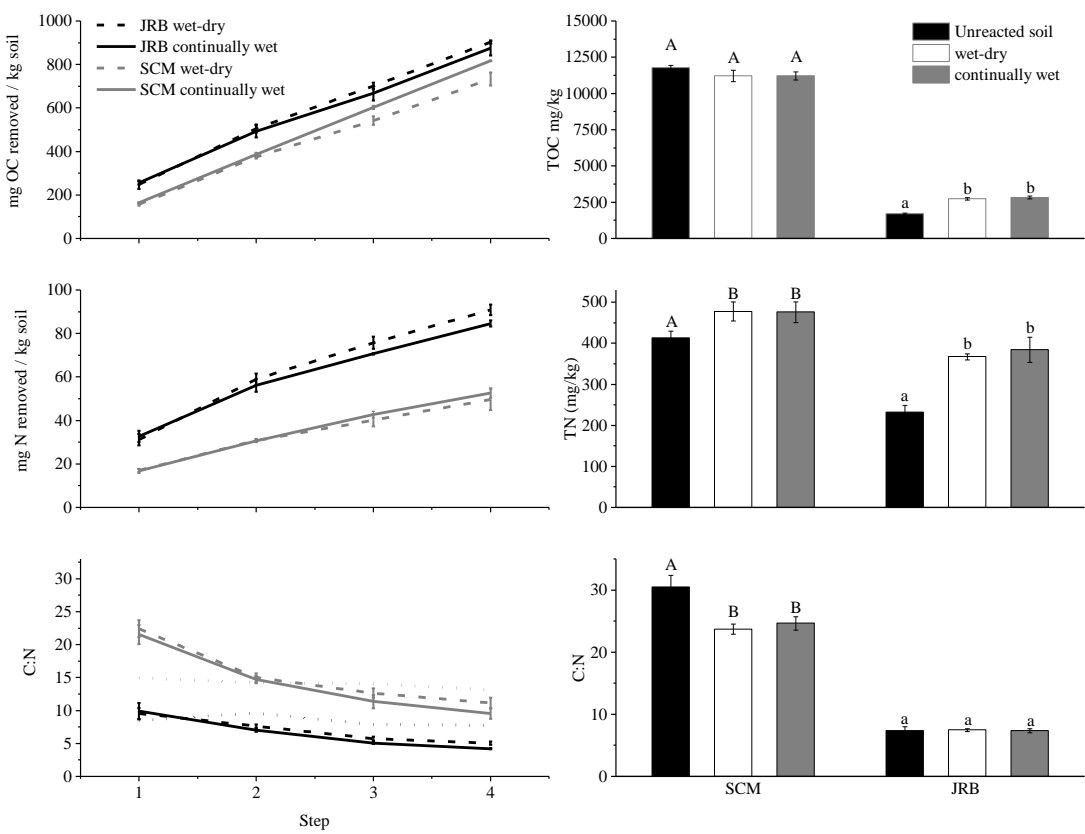



**Figure 1: The organic carbon (top), nitrogen (middle) and C:N (bottom), for equilibrated solutions (left) and solid phases after four reaction steps (right). Values for equilibrated solution OC and N represent cumulative removal from solution per soil mass. Dashed lines in OC and N plots show continuous-wet treatments, dotted lines in the C:N plot represent values of unreacted DOM solutions, error bars are the standard deviation, and letters indicate significant difference ($p<0.05$) from unreacted control.**

509

510

511




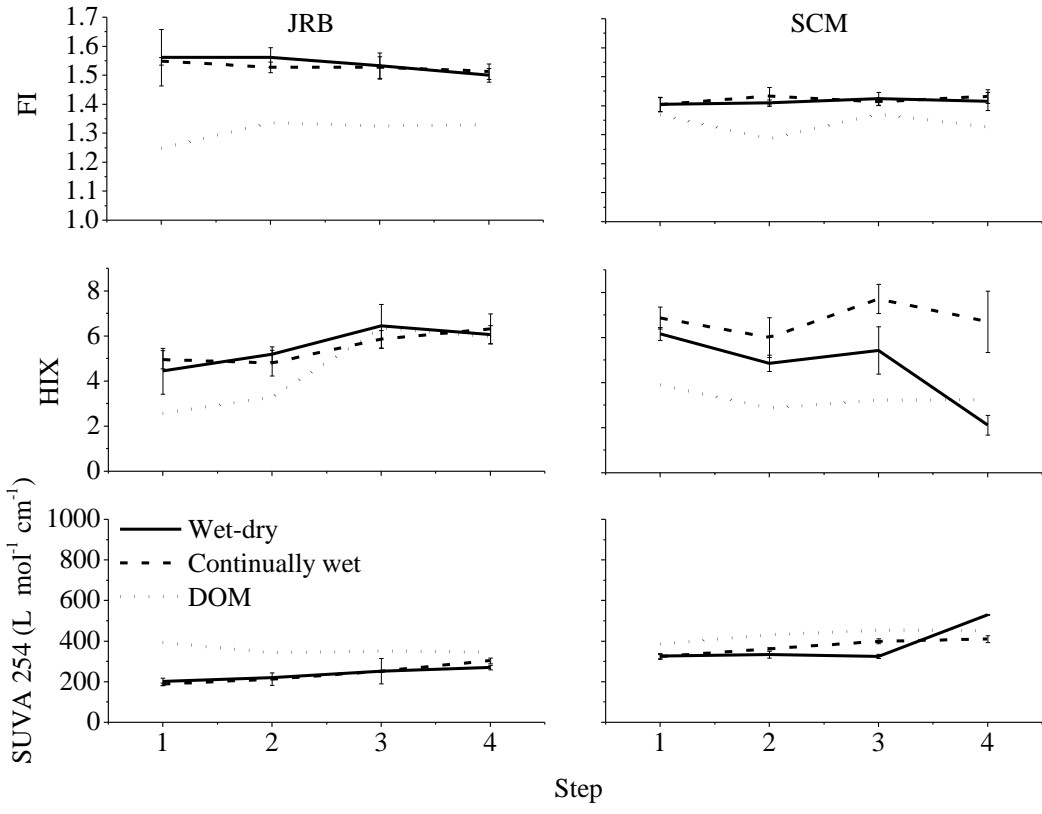

**Figure 2. The fluorescence Index (FI), humification index (HIX) and specific UV absorbance at 245 nm (SUVA$_{254}$), for equilibrated solutions reacted with JRB and SCM soils. The solid lines are wet-dry series, dashed lines are continuous-wet, and dotted lines are unreacted DOM; error bars are the standard deviation.**





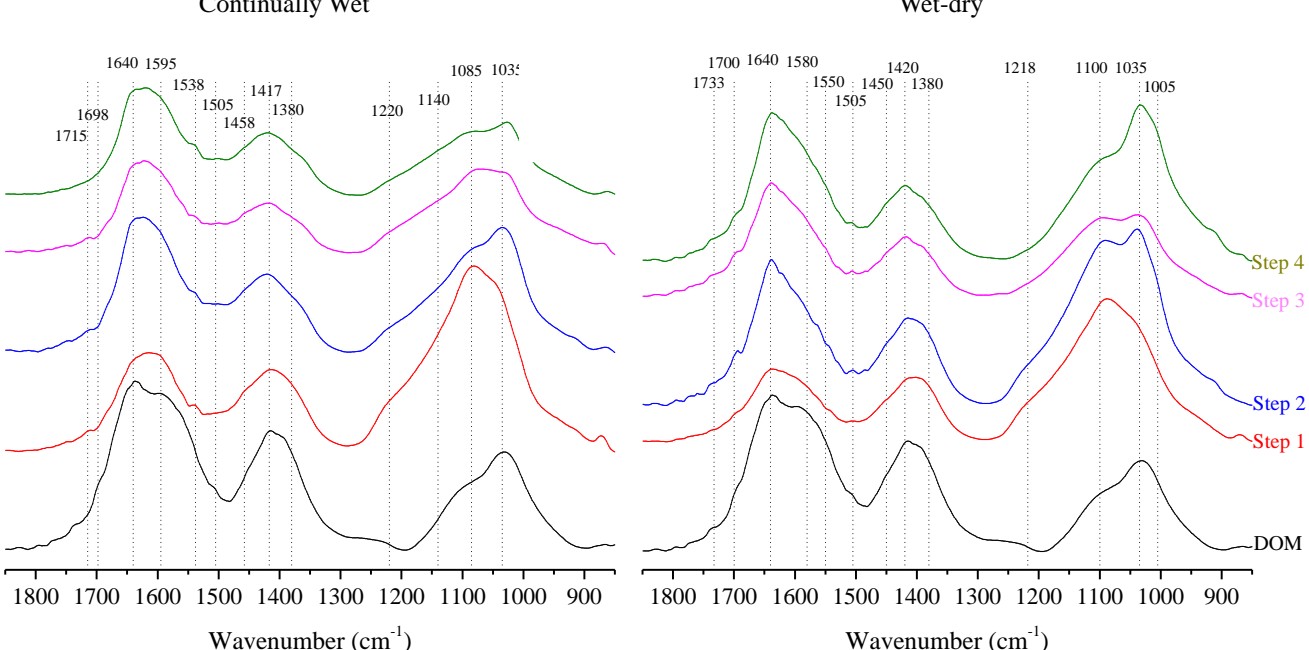

**Figure 3. Transmission FTIR spectra of the DOM dried solution reacted with JRB soils from steps 1 to 4 for continuous-wet (left) and wet-dry cycled (right) and the unreacted JRB DOM solution (bottom black line). For color rendering of this image please refer to the online version.**









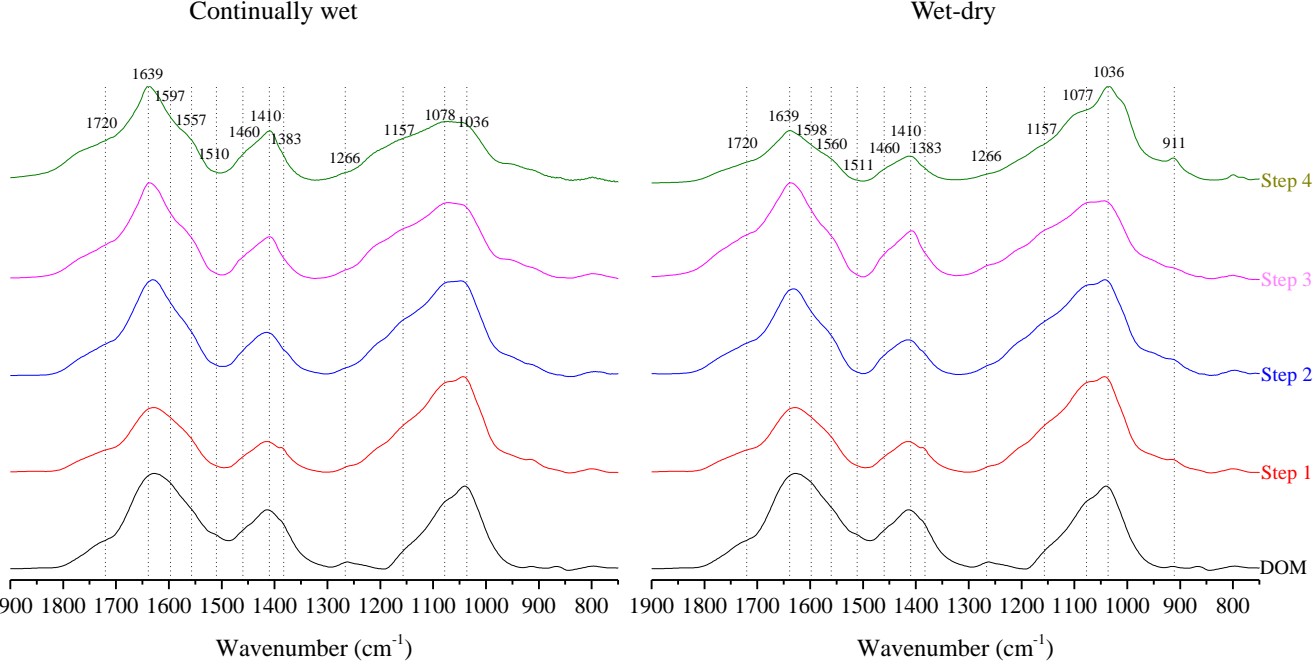

Figure 4. Transmission FTIR spectra of the DOM dried solution reacted with SCM soils from steps 1 to 4 for continuous-wet (left) and wet-dry cyclied (right) and the unreacted SCM DOM solution (bottom line). For color rendering of this image please refer to the online version.





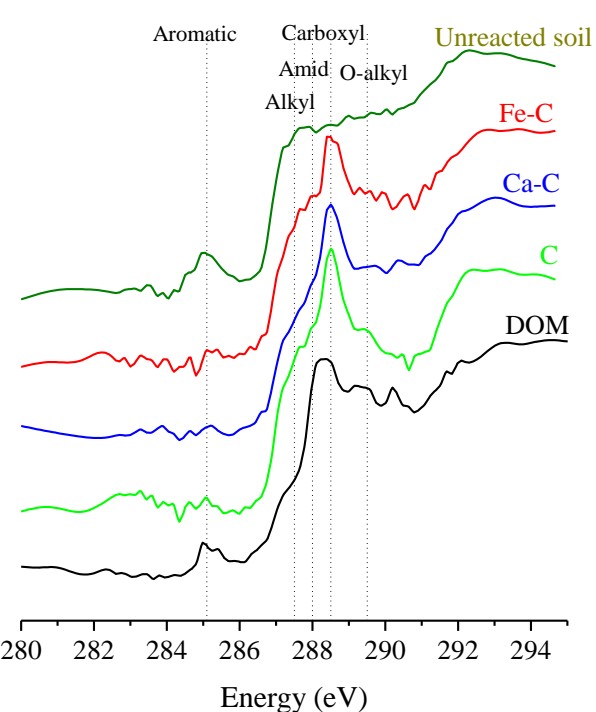
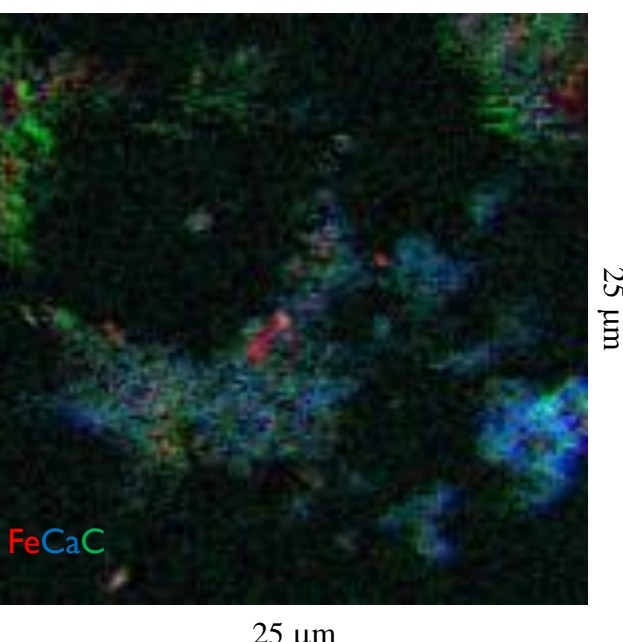



**Figure 5. JRB soil reacted with DOM under wet-dry cycling. Left, C NEXAFS spectra extracted from C, Ca, and Fe regions of STXM map. Spectra of unreacted soil (top) and DOM solution (bottom) are presented. Dashed vertical lines point out C species. Right, tri-colored STXM map of fine fraction from JRB soil reacted four times with DOM under wet-dry cycling; Fe (red), Ca (blue) and C (green). Image size 25 x 25 μm. For color rendering of this image please refer to the online version.**

535



536

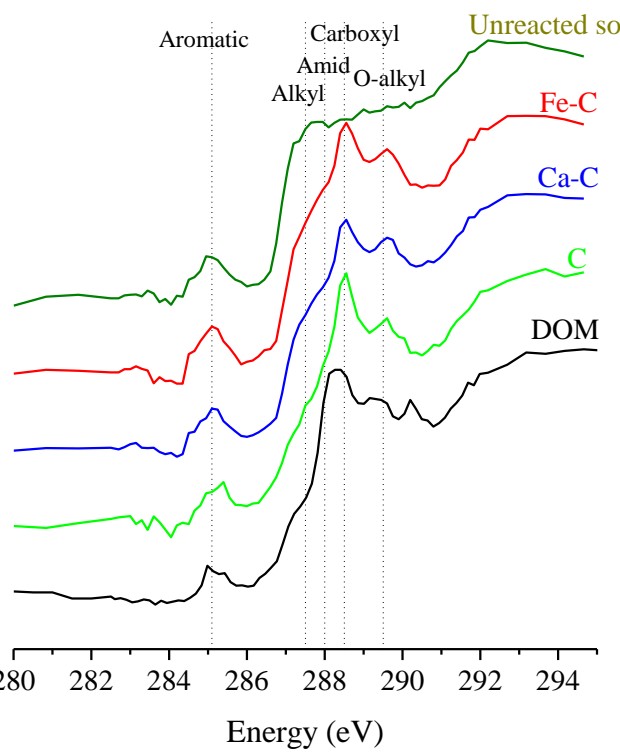

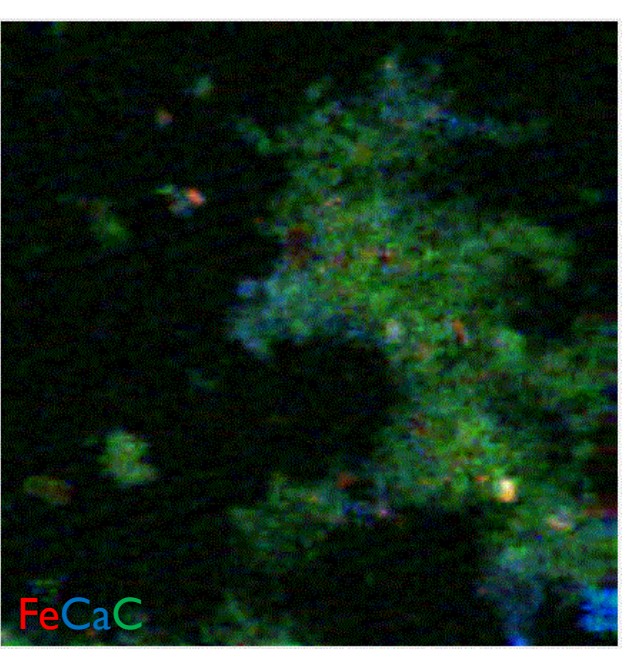

25 μm

25 μm

**Figure 6. JRB soil reacted with DOM under continuous-wet conditions. Left, C NEXAFS spectra extracted from C, Ca, and Fe regions of STXM map. Spectra of unreacted soil (top) and DOM solution (bottom) are presented. Dashed vertical lines point out C species. Right, tri-colored STXM map of fine fraction from JRB soil reacted four times with DOM during the continuous-wet treatment. Fe (red), Ca (blue) and C (green). Image size 25 x 25 μm. For color rendering of this image please refer to the online version.**

542





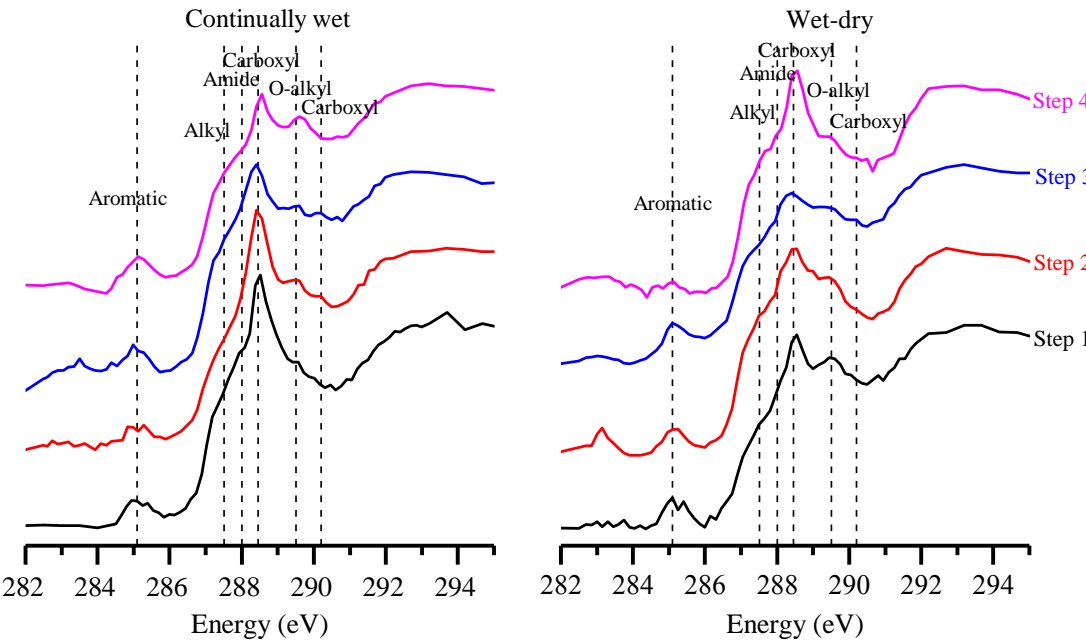

**Figure 7. C NEXAFS extracted from C (red in Fig 6) regions of STXM map for the second step of the continuous-wet treatment (left) and from all 4 steps of the wet-dry treatment (right). For color rendering of this image please refer to the online version.**