# Peer review of "Wet-dry cycles impact DOM retention in subsurface soils"

_Biogeosciences, 2017_

## Referee Comment (RC1) · Anonymous Referee #1 · 26 Sep 2017

This manuscript describes the effect of wet-dry cycles on DOM retention and composition in two subsurface soils. The manuscript is very well written and the description of the experiment and results are clear and sound.

Overall, this manuscript provides valuable insights in the effect of wet-dry cycles on DOM retention and the nm-scale interactions between DOM and mineral soil and is of interest to the scientific community and readers of Biogeosciences in particular.

Specific Comments: - In the methods and results section it is not really stated clearly that an active microbial community was present while the experiment proceeded. As is discussed line 297-311, page 11, microbial decomposition of DOC was clearly relevant. This should be stated more clearly in the method and result section.

- In addition, it would be very helpful to the reader if a carbon mass balance or estimation of the % carbon lost due to mineralization was reported

- It is reported that the OC content of SCM was effectively unchanged after treatment (line 195). However, given the large error margin on the solid phase OC content measurement, it is also possible the change in OC content due to DOC retention falls within the margin of error of the measurement. The much larger relative accumulation of DOC in the solid phase of JRB is more easily detected, but judging from figure 1, the OC removal from solution behaves similarly. Unless additional evidence is provided to demonstrate a higher microbial biomass, activity and/or higher carbon mineralization, I am not convinced it can be stated that enhanced mineralization took place in SCM relative to JRB (line 303) based on the data presented here.

- The effect of mineralization and/or transformation of DOC during the experiment could have a major impact on the results of both quantity and composition of the OC retained in the soil. In addition, the duration of the experiment (4 times 98 hours) is sufficient for microbial growth to take place in response to the DOC addition and treatment. This should be addressed carefully in the discussion, particularly also in relation to the difference between the continually wet and wet-dry treatments.

- It is stated quite strongly in the abstract (line 20) and conclusions (line 354-356) that the spatial fractionation of adsorbed OM is different from what was expected based on previous literature. Though I agree that this is a very interesting observation, a more detailed evaluation of these observations in relation to previous literature is missing and should be added to the discussion.

- Line 290-296: it could be of interest here to also note the change in ionic strength of the pore-water that occurs due to air drying of the soil and its possible effect on the local Ca2+ concentration and thereby potential to form complexes with OM

---

## Referee Comment (RC2) · Anonymous Referee #2 · 30 Sep 2017

General comments The paper is well written and structured and provides new insights in the retention of dissolved organic matter in soil under varying moisture conditions, which is highly interesting especially for the scientific community dedicated to elucidate organic carbon dynamics along the soil profile. The chosen methodological tool set is innovative and suitable to reach the proposed research aims. I have some comments that should be addressed before the paper is potentially acceptable for publication.

Specific comments The entire paper needs to be checked for a proper introduction of the used abbreviation and their constant usage throughout the paper. Lines 15-19: split into two sentences Line 24: "considered" not "consider" Line 27: introduce the abbreviation for carbon and then use this abbreviation constantly throughout the paper Lines 68-71: give some explanations why do you expect an increase in surface complexation

of carboxyl groups with metal (oxy)hydroxide surfaces or hydroxylated edge surfaces of aluminosilicate clays and promoted associations of hydrophobic fractions with pre-adsorbed and desiccated DOM components especially in the light of the relatively high pH value of the JRB soil and the statement in lines 348-349 that it is important to note that the studied soils are predominantly composed of silicate and aluminosilicate minerals and are relatively depleted in crystalline and short range order metal oxides Lines 86-89: explain the sampling design in more detail, at the moment the reader is not able to understand if true field replicates, pseudo field replicates or just laboratory replicates of one composite sample were analysed, this is important in relation to the statistical tests that were performed, which require at least pseudo field replicates from, for example, three different soil pits per site, also I did not find in the text any information about the number of replicates Lines 109-111: the 28 ml that have been filtered, is this the decanted aliquot or did this volume was removed by pipetting? please explain in more detail Lines 133-134: why did you use different preparation techniques for the two soils, and how this affects the comparability of the spectra? Line 250: for better understanding, please explain "relative", did you compute ratios or normalized the spectra to the o-alkyl peak? Lines 303-305: this statement has to be backed up with references and including a discussion of information from the literature because in this study no direct measurements of microbial parameter were carried out Lines 310-311: any recommendation how to address this? how should an experiment be designed to get further information here 354-356: this criticism is stressed specifically in the Abstract and in the Conclusions, if the authors wish to make a point here, they should discuss this in more detail including some recommendation for future experimental work

---

## Author Comment (AC1) · 4 Nov 2017

We are grateful for the Referees' comments on our submission; addressing them will certainly improve the paper. We are hereby uploading our numbered responses to referee comments (RCs) as well as our indications of how we plan to implement the suggested changes.

Please find below our respective response to each RC. We will add detail (and make cuts where needed) for improved clarity in the final revision.

Anonymous Referee #1

This manuscript describes the effect of wet-dry cycles on DOM retention and composition in two subsurface soils. The manuscript is very well written and the description

of the experiment and results are clear and sound. Overall, this manuscript provides valuable insights in the effect of wet-dry cycles on DOM retention and the nm-scale interactions between DOM and mineral soil and is of interest to the scientific community and readers of Biogeosciences in particular.

Specific Comments:

RC 1: In the methods and results section, it is not really stated clearly that an active microbial community was present while the experiment proceeded. As is discussed line 297-311, page 11, microbial decomposition of DOC was clearly relevant. This should be stated more clearly in the method and result section.

Response 1: We agree with the referee, in this study microbial activity was not suppressed. The revised version will include better clarification on this point.

RC 2: In addition, it would be very helpful to the reader if a carbon mass balance or estimation of the % carbon lost due to mineralization was reported

Response 2: This is a good point. The estimated mass of carbon that was potentially mineralized in the SCM soil systems were, 1370 $\pm$ 840 and 1440 $\pm$ 680 mg organic carbon (OC) kg-1 soil (for wet-dry and continuous-wet treatments respectively). These values represents 11 $\pm$ 7 and 11 $\pm$ 5 % of the total carbon in the wet-dry and continually-wet systems. These mass balance values will be added to the revised version.

RC 3: It is reported that the OC content of SCM was effectively unchanged after treatment (line 195). However, given the large error margin on the solid phase OC content measurement, it is also possible the change in OC content due to DOC retention falls within the margin of error of the measurement. The much larger relative accumulation of DOC in the solid phase of JRB is more easily detected, but judging from figure 1, the OC removal from solution behaves similarly. Unless additional evidence is provided to demonstrate a higher microbial biomass, activity and/or higher carbon mineralization,

[Figure]

I am not convinced it can be stated that enhanced mineralization took place in SCM relative to JRB (line 303) based on the data presented here.

Response 3: To address the referee's comment we performed additional parametric statistical analysis for the OC, TN and C to N data. ANOVA and Tukey's HSD were used to test the difference between means. We then compered the mean loss of OC in the reacted soils and the mean amount of OC removed from solution with the student t-test. Results demonstrate a significant mass loss of OC from SCM soil system ($p \leq 0.05$), amounting to $1370 \pm 840$ and $1440 \pm 680$ mg OC kg-1 soil (for wet-dry and continuous-wet treatments respectively). In the JRB system, the total mass differences were not statistically significant ($p > 0.95$) $150 \pm 230$ and $260 \pm 250$ mg OC kg-1 soil (for wet-dry and continuous-wet treatments respectively). In addition, C to N ratio of reacted SCM soils significantly decreased compared to unreacted soil, suggesting microbial transformation of the soil organic matter. Therefore, although this study did not measures microbial biomass or activity directly, our result do support OC mineralization in the SCM soil case. This additional data, analysis and discussion will be added to the revised version.

RC 4: The effect of mineralization and/or transformation of DOC during the experiment could have a major impact on the results of both quantity and composition of the OC retained in the soil. In addition, the duration of the experiment (4 times 98 hours) is sufficient for microbial growth to take place in response to the DOC addition and treatment. This should be addressed carefully in the discussion, particularly also in relation to the difference between the continually wet and wet-dry treatments.

Response 4: We agree with this RC that microbial growth and respiration contribute to mineralization and transformation of DOC. Our results indicate that OC decomposition is statistically significant in terms of carbon mass balance in the SCM system, whereas it is not clearly detectable above experimental error in the JRB system (see responses 2 and 3 above). Relative increases in peak intensity of O-alkyl from FTIR spectra of the wet-dry reacted SCM soil suggest an increased contribution of microbial derived material. However, spectroscopy (FTIR) and micro-spectroscopy (STXM-NEXAFS) analysis of reacted JRB DOM solutions and soils, suggest that the functional groups removed from reacted JRB DOM solution were similar to the ones accumulated in the reacted JRB soils; therefore, on this basis and the carbon balance, we conclude that the reactions were dominated by sorption and not biotransformation. The revised version will contain more specific discussion of this point.

RC 5: It is stated quite strongly in the abstract (line 20) and conclusions (line 354-356) that the spatial fractionation of adsorbed OM is different from what was expected based on previous literature. Though I agree that this is a very interesting observation, a more detailed evaluation of these observations in relation to previous literature is missing and should be added to the discussion.

Response 5: The referee makes an important point that warrants additional information and reference to the literature in the discussion. As has been summarized in the Introduction section (lines 51-65), previous studies demonstrated the important of short-range order metal (oxy)hydroxides for the selective adsorption of phenolic and carboxylic moieties, whereas layer silicate clay minerals are shown to preferentially adsorb aliphatic OM (Chorover and Amistadi, 2001). The present study used for the first time STXM-NEXAFS micro-spectroscopy to test if such fractionation occurs on the nano-scale of heterogeneous soil surfaces. Our results demonstrate that although distinct soil surfaces were enriched with iron (i.e. (oxy)hydroxides), the OC associated with these surfaces did not differ detectably from the bulk OC. We will emphasize this point with reference to the literature in the discussion section of the revised version.

RC 6: Line 290-296: it could be of interest here to also note the change in ionic strength of the pore-water that occurs due to air drying of the soil and its possible effect on the local $Ca^{2+}$ concentration and thereby potential to form complexes with OM

Response 6: This is a valid point as it has previously been shown that ionic strength may indeed affect complexion of OM with metal cations and particle surfaces. The

revised version will include discussion of the importance of soil solution ionic strength.

Anonymous Referee #2

General comments: The paper is well written and structured and provides new insights in the retention of dissolved organic matter in soil under varying moisture conditions, which is highly interesting especially for the scientific community dedicated to elucidate organic carbon dynamics along the soil profile. The chosen methodological tool set is innovative and suitable to reach the proposed research aims. I have some comments that should be addressed before the paper is potentially acceptable for publication.

Specific comments: RC 1: The entire paper needs to be checked for a proper introduction of the used abbreviation and their constant usage throughout the paper.

Response 1: We thank the referee for addressing this point, the revised version will have proper introductions of all abbreviations.

RC 2: Lines 15-19: split into two sentences

Response 2: We agree with the referee and will change the sentence accordingly.

RC 3: Line 24: "considered" not "consider"

Response 3: Change will be made.

RC 4: Line 27: introduce the abbreviation for carbon and then use this abbreviation constantly throughout the paper

Response 4: The new version will include the correct and consistent abbreviation for carbon.

RC 5: Lines 68-71: give some explanations why do you expect an increase in surface complexation of carboxyl groups with metal (oxy)hydroxide surfaces or hydroxylated edge surfaces of aluminosilicate clays and promoted associations of hydrophobic fractions with preadsorbed and desiccated DOM components especially in the light of the

relatively high pH value of the JRB soil and the statement in lines 348-349 that it is important to note that the studied soils are predominantly composed of silicate and aluminosilicate minerals and are relatively depleted in crystalline and short-range order metal oxides.

Response 5: Based on previous work that tested the interactions natural organic matter and reference mineral phases, carboxyl groups demonstrate a higher reactivity toward metal (oxy)hydroxide surfaces or hydroxylated edge surfaces of aluminosilicates than they do for other prevalent mineral surfaces in soils (e.g., charged or uncharged basal plane siloxane sites of clay minerals). The relevant literature is summarized in the introduction section (lines 51-65). In the current study, a combination of STXM-NEXAFS analysis was used to visualize this phenomenon at the sub-micron scale on natural soil surfaces. The results demonstrate that although distinct soil surfaces were enriched with iron (i.e. (oxy)hydroxides), the OC associated with these surfaces did not differ detectably from the bulk OC. We will clarify the context of this observation in light of prior work in the revised version.

RC 6: Lines 86-89: explain the sampling design in more detail, at the moment the reader is not able to understand if true field replicates, pseudo field replicates, or just laboratory replicates of one composite sample were analysed, this is important in relation to the statistical tests that were performed, which require at least pseudo field replicates from, for example, three different soil pits per site, also I did not find in the text any information about the number of replicates.

Response 6: In this study, soil samples were composited during collection from a single pit excavated in each site. The samples were collected from different locations within each pit and composited to one representative local sample. Replication therefore pertains to the experiment itself, i.e., to test the reproducibility and uncertainty associated with the effects of the wet-dry and continuous-wet treatments for a given soil type. This additional information on sample collection and experimental replication will be added to the revised version.

RC 7: Lines 109-111: the 28 ml that have been filtered, is this the decanted aliquot or did this volume was removed by pipetting? Please explain in more detail.

Response 7: The supernatant was removed by careful pipetting just below the surface to avoid loss of solids. Details will be added to the revised version.

RC 8: Lines 133-134: why did you use different preparation techniques for the two soils, and how this affects the comparability of the spectra?

Response 8: Despite the variation mentioned above, it is first most important to point out that transmission FTIR spectroscopy was used in all cases. Also, although not stated in the earlier version of the manuscript, OM samples were subjected to both transmission sample introduction protocols, the KBr pellet and infrared transmission window approach, to provide a direct comparison from which we selected the datasets most informative on the treatment effects within a given soil. Air drying DOM onto IR transmission (Ge or ZnSe) windows is generally considered the approach least prone to artifacts associated with freeze-drying (required for KBr pellets) and potential chemical reaction with the KBr pellet during pressurized preparation (Johnston and Aochi, 1996). Conversely, the potential shortcoming of this method is that spectra may have a lower signal to noise ratio than other sample introduction methods. In the case of the JRB soil, despite high signal to noise for the KBr spectra, we observed alteration of vibrations that suggested potential reaction with the KBr matrix upon pelletization. This type of matrix interaction with DOM is possible and needs to be evaluated on a case-by-case basis (Johnston and Aochi, 1996). Since alteration of spectra in KBr was apparent and transmission window spectra were of high quality, the transmission window data were included as the best available dataset to evaluate the changes with time and treatment for this DOM. Conversely, no alteration of spectra in KBr was observed for SCM soil. Therefore the higher quality KBr pellet spectra are shown for the SCM soil. Most importantly, our FTIR spectral comparisons are focused on characterization of molecular changes that occurred within a given soil by treatment. They were not used to make quantitative comparisons between soils. Hence the sample introduction

methods, although different between samples, were consistent across all treatments within a given soil, and so do not alter the inferences made in terms of the effects of treatment and time.

RC 9: Line 250: for better understanding, please explain "relative", did you compute ratios or normalized the spectra to the O-alkyl peak?

Response 9: Changes in solution composition of OC were evaluated by the ratio between normalized peak intensity in the FTIR spectra. Clarification will be added to the revised version.

RC 10: Lines 303-305: this statement has to be backed up with references and including a discussion of information from the literature because in this study no direct measurements of microbial parameter were carried out

Response 10: We agree with the RC. The revised version will include additional literature and discussion to support this point.

RC 11: Lines 310-311: any recommendation how to address this? How should an experiment be designed to get further information here?

Response 11: Experimental design using isotopically labeled material would provide additional information regarding exchange and decomposition reactions.

RC 12: Line 354-356: this criticism is stressed specifically in the Abstract and in the Conclusions, if the authors wish to make a point here, they should discuss this in more detail including some recommendation for future experimental work.

Response 12: Excellent point. As mentioned in the response to comment 5, additional discussion will be added to the revised version. Future experiments using soils with higher relative proportions of short-range order metal (oxy)hydroxide and lower organic carbon content may provide additional information on nano-scale spatial fractionation of OC.

References: Chorover, J. and Amistadi, M. K.: Reaction of forest floor organic matter at goethite, birnessite and smectite surfaces, Geochim. Cosmochim. Acta, 65(1), 95–109, 2001. Johnston, C. T., and Y. O. Aochi. 1996. Fourier transform infrared and Raman spectroscopy. In D. L. Sparks (ed.) Methods of Soil Analysis. Part 3. Chemical Methods-SSSA Book Series no. 5. Chapter 10, pp. 269-321. Soil Science Society of America, Madison, WI.

—————————————————————

---

## Author Response (AR1)

**Responses to Referee comments (RCs) for "Wet-dry cycles impact DOM retention in subsurface soils" by Yaniv Olshansky et al.**

Dear Associate Editor and referees,

We are grateful for the Referees' and your own comments on our submission; addressing them has certainly improved the paper. We are hereby uploading our numbered responses to referee comments (RCs) as well as our indications of how we implemented the suggested changes.

Please find below our respective response to each RC in *blue*. We have added detail (and made cuts where needed) for improved clarity in this final revision.

**Anonymous Referee #1 (R1)**

This manuscript describes the effect of wet-dry cycles on DOM retention and composition in two subsurface soils. The manuscript is very well written and the description of the experiment and results are clear and sound. Overall, this manuscript provides valuable insights in the effect of wet-dry cycles on DOM retention and the nm-scale interactions between DOM and mineral soil and is of interest to the scientific community and readers of Biogeosciences in particular.

**Specific Comments:**

**R1C 1:** In the methods and results section, it is not really stated clearly that an active microbial community was present while the experiment proceeded. As is discussed line 297-311, page 11, microbial decomposition of DOC was clearly relevant. This should be stated more clearly in the method and result section.

**Response 1:** We agree with the referee, in this study microbial activity was not suppressed. The revised version includes better clarification on this point. (lines 121-122 and 312-315)

**R1C 2:** In addition, it would be very helpful to the reader if a carbon mass balance or estimation of the % carbon lost due to mineralization was reported

**Response 2:** This is a good point. The estimated mass of carbon that was potentially mineralized in the SCM soil systems were, $1370 \pm 840$ and $1440 \pm 680$ mg organic carbon (OC) $kg^{-1}$ soil (for wet-dry and continuous-wet treatments respectively). These values represent $11 \pm 7$ and $11 \pm 5$ % of the total carbon in the wet-dry and continually-wet systems. These mass balance values were added to the revised version. (lines 202-206)

**R1C 3:** It is reported that the OC content of SCM was effectively unchanged after treatment (line 195). However, given the large error margin on the solid phase OC content measurement, it is also possible the change in OC content due to DOC retention falls within the margin of error of the measurement. The much larger relative accumulation of DOC in the solid phase of JRB is more easily detected, but judging from figure 1, the OC removal from solution behaves similarly. Unless additional evidence is provided to demonstrate a higher microbial biomass, activity and/or higher carbon mineralization, I am not convinced it can be stated that enhanced mineralization took place in SCM relative to JRB (line 303) based on the data presented here.

**Response 3:** To address the referee's comment we performed additional parametric statistical analysis for the OC, TN and C to N data. ANOVA and Tukey's HSD were used to test the difference between means. We then compered the mean loss of OC in the reacted soils and the mean amount of OC removed from solution with the student t-test. Results demonstrated a significant mass loss of OC from SCM soil system ($p \leq 0.05$), amounting to $1370 \pm 840$ and $1440 \pm 680$ mg OC $kg^{-1}$ soil (for wet-dry and continuous-wet treatments respectively), added at line 204. In the JRB system, the total mass differences were not statistically significant ($p > 0.95$) $150 \pm 230$ and $260 \pm 250$ mg OC $kg^{-1}$ soil (for wet-dry and continuous-wet treatments respectively). In addition, C to N ratio of reacted SCM soils significantly decreased compared to unreacted soil, suggesting microbial transformation of the soil organic matter. Therefore, although this study did not measures microbial biomass or activity directly, our result do support OC mineralization in the SCM soil case. This additional data, analysis and discussion was added to the revised version. (lines 163-166, 202-206 and caption of figure 1)

**R1C 4:** The effect of mineralization and/or transformation of DOC during the experiment could have a major impact on the results of both quantity and composition of the OC retained in the soil. In addition, the duration of the experiment (4 times 98 hours) is sufficient for microbial growth to take place in response to the DOC addition and treatment. This should be addressed carefully in the discussion, particularly also in relation to the difference between the continually wet and wet-dry treatments.

**Response 4:** We agree with this RC that microbial growth and respiration contribute to mineralization and transformation of DOC. Our results indicate that OC decomposition is statistically significant in terms of carbon mass balance in the SCM system, whereas it is not clearly detectable above experimental error in the JRB system (see responses 2 and 3 above). Relative increases in peak intensity of O-alkyl from FTIR spectra of the wet-dry reacted SCM soil suggest an increased contribution of microbial derived material. However, spectroscopy (FTIR) and micro-spectroscopy (STXM-NEXAFS) analysis of reacted JRB DOM solutions and soils, suggest that the functional groups removed from reacted JRB DOM solution were similar to the ones accumulated in the reacted JRB soils; therefore, on this basis and the carbon balance, we conclude that the reactions were dominated by sorption and not biotransformation. The revised version contain a more specific discussion of this point. (line 312-315)

**R1C 5:** It is stated quite strongly in the abstract (line 20) and conclusions (line 354-356) that the spatial fractionation of adsorbed OM is different from what was expected based on previous literature. Though I agree that this is a very interesting observation, a more detailed evaluation of these observations in relation to previous literature is missing and should be added to the discussion.

**Response 5:** The referee makes an important point that warrants additional information and reference to the literature in the discussion. As has been summarized in the Introduction section (lines 51-65), previous studies demonstrated the importance of short-range order metal (oxy)hydroxides for the selective adsorption of phenolic and carboxylic moieties, whereas layer silicate clay minerals are shown to preferentially adsorb aliphatic OM (Chorover and Amistadi, 2001). The present study used for the first time STXM-NEXAFS micro-spectroscopy to test if such fractionation occurs on the nano-scale of heterogeneous soil surfaces. Our results demonstrate that although distinct soil surfaces were enriched with iron (i.e. (oxy)hydroxides), the OC associated with these surfaces did not differ detectably from the bulk OC. We emphasized this point with reference to the literature in the discussion section of the revised version. (lines 351-355)

**R1C 6:** Line 290-296: it could be of interest here to also note the change in ionic strength of the pore-water that occurs due to air drying of the soil and its possible effect on the local $Ca^{2+}$ concentration and thereby potential to form complexes with OM.

**Response 6:** This is a valid point as it has previously been shown that ionic strength may indeed affect complexion of OM with metal cations and particle surfaces. The revised version will include discussion of the importance of soil solution ionic strength. (lines 307-308)

**Anonymous Referee #2 (R2)**

General comments: The paper is well written and structured and provides new insights in the retention of dissolved organic matter in soil under varying moisture conditions, which is highly interesting especially for the scientific community dedicated to elucidate organic carbon dynamics along the soil profile. The chosen methodological tool set is innovative and suitable to reach the proposed research aims. I have some comments that should be addressed before the paper is potentially acceptable for publication. Specific comments:

**R2C 1:** The entire paper needs to be checked for a proper introduction of the used abbreviation and their constant usage throughout the paper.

**Response 1:** We thank the referee for addressing this point, the revised version has the proper introductions of all abbreviations.

**R2C 2:** Lines 15-19: split into two sentences

**Response 2:** We agree with the referee and changed the sentence accordingly.

**R2C 3:** Line 24: "considered" not "consider"

**Response 3:** This was corrected.

**R2C 4:** Line 27: introduce the abbreviation for carbon and then use this abbreviation constantly throughout the paper

**Response 4:** The new version includes the correct and consistent abbreviation for carbon throughout.

**R2C 5:** Lines 68-71: give some explanations why do you expect an increase in surface complexation of carboxyl groups with metal (oxy)hydroxide surfaces or hydroxylated edge surfaces of aluminosilicate clays and promoted associations of hydrophobic fractions with preadsorbed and desiccated DOM components especially in the light of the relatively high pH value of the JRB soil and the statement in lines 348-349 that it is important to note that the studied soils are predominantly composed of silicate and aluminosilicate minerals and are relatively depleted in crystalline and short range order metal oxides.

**Response 5:** Based on previous work that tested the interactions natural organic matter and reference mineral phases, carboxyl groups demonstrate a higher reactivity toward metal (oxy)hydroxide surfaces or hydroxylated edge surfaces of aluminosilicates than they do for other prevalent mineral surfaces in soils (e.g., charged or uncharged basal plane siloxane sites of clay minerals). The relevant literature is summarized in the introduction section (lines 51-65). In the current study, a combination of STXM-

NEXAFS analysis was used to visualize this phenomenon at the sub-micron scale on natural soil surfaces. The results demonstrate that although distinct soil surfaces were enriched with iron (i.e. (oxy)hydroxides), the OC associated with these surfaces did not differ detectably from the bulk OC. We have clarified the context of this observation in light of prior work in the revised version. (lines 351-355)

**R2C 6:** Lines 86-89: explain the sampling design in more detail, at the moment the reader is not able to understand if true field replicates, pseudo field replicates, or just laboratory replicates of one composite sample were analysed, this is important in relation to the statistical tests that were performed, which require at least pseudo field replicates from, for example, three different soil pits per site, also I did not find in the text any information about the number of replicates.

**Response 6:** In this study, soil samples were composited during collection from a single pit excavated in each site. The samples were collected from different locations within each pit and composited to one representative local sample. Replication therefore pertains to the experiment itself, i.e., to test the reproducibility and uncertainty associated with the effects of the wet-dry and continuous-wet treatments for a given soil type. This additional information on sample collection and experimental replication has been added to the revised version. (lines 86-89)

**R2C 7:** Lines 109-111: the 28 ml that have been filtered, is this the decanted aliquot or did this volume was removed by pipetting? Please explain in more detail.

**Response 7:** The supernatant was removed by careful pipetting just below the surface to avoid loss of solids. Details have been added to the revised version. (line 111-113)

**R2C 8:** Lines 133-134: why did you use different preparation techniques for the two soils, and how this affects the comparability of the spectra?

**Response 8:** Despite the variation mentioned above, it is first most important to point out that transmission FTIR spectroscopy was used in all cases. Also, although not stated in the earlier version of the manuscript, OM samples were subjected to both transmission sample introduction protocols, the KBr pellet and infrared transmission window approach, to provide a direct comparison from which we selected the datasets most informative on the treatment effects within a given soil. Air drying DOM onto IR transmission (Ge or ZnSe) windows is generally considered the approach least prone to artifacts associated with freeze-drying (required for KBr pellets) and potential chemical reaction with the KBr pellet during pressurized preparation (Johnston and Aochi, 1996). Conversely, the potential shortcoming of this method is that spectra may have a lower signal to noise ratio than other sample introduction methods. In the case of the JRB soil, despite high signal to noise for the KBr spectra, we observed alteration of vibrations that suggested potential reaction with the KBr matrix upon pelletization. This type of matrix interaction with DOM is possible and needs to be evaluated on a case-by-case basis (Johnston and Aochi, 1996). Since alteration of spectra in KBr was apparent and transmission window spectra were of high quality, the transmission window data were included as the best available dataset to evaluate the changes with time and treatment for this DOM. Conversely, no alteration of spectra in KBr was observed for SCM soil. Therefore the higher quality KBr pellet spectra are shown for the SCM soil. Most importantly, our FTIR spectral comparisons are focused on characterization of molecular changes that occurred within a given soil by treatment. They were not used to make quantitative comparisons between soils. Hence the sample introduction methods, although different between samples, were consistent across all treatments within a given soil, and so do not alter the inferences made in terms of the effects of treatment and time.

**R2C 9:** Line 250: for better understanding, please explain "relative", did you compute ratios or normalized the spectra to the O-alkyl peak?

**Response 9:** Changes in solution composition of OC were evaluated by the ratio between normalized peak intensity in the FTIR spectra. Clarification has been added to the revised version. (lines 177-178)

**R2C 10:** Lines 303-305: this statement has to be backed up with references and including a discussion of information from the literature because in this study no direct measurements of microbial parameter were carried out

**Response 10:** We agree with the RC. The revised version includes additional literature citation and discussion to support this point. (lines 316-320)

**R2C 11:** Lines 310-311: any recommendation how to address this? How should an experiment be designed to get further information here?

**Response 11:** Experimental design using isotopically labeled material would provide additional information regarding exchange and decomposition reactions, we have made a note of this (line 326-328).

**R2C 12:** Line 354-356: this criticism is stressed specifically in the Abstract and in the Conclusions, if the authors wish to make a point here, they should discuss this in more detail including some recommendation for future experimental work.

**Response 12:** Excellent point. As mentioned in the response to comment 5, additional discussion have been added to the revised version. Future experiments using soils with higher relative proportions of short-range order metal (oxy)hydroxide and lower organic carbon content may provide additional information on nano-scale spatial fractionation of OC, and this has been added to the conclusion. (lines 377-378)

References added:

Chorover, J. and Amistadi, M. K.: Reaction of forest floor organic matter at goethite, birnessite and smectite surfaces, Geochim. Cosmochim. Acta, 65(1), 95–109, 2001.

Johnston, C. T., and Y. O. Aochi. 1996. Fourier transform infrared and Raman spectroscopy. In D. L. Sparks (ed.) Methods of Soil Analysis. Part 3. Chemical Methods-SSSA Book Series no. 5. Chapter 10, pp. 269-321. Soil Science Society of America, Madison, WI.

[revised manuscript text omitted]

---

## Author Response (AR2)

**Responses to Referee comments for "Wet-dry cycles impact DOM retention in subsurface soils" by Yaniv Olshansky et al.**

Dear Associate Editor and referees,

We are grateful for the Referees' and your own comments on our submission; addressing them has certainly improved the paper. We are hereby uploading our responses to referee comments as well as our revised version. All changes are highlight in yellow.

Referee comments:

The manuscript has been much improved, and referee comments have been addressed appropriately.

I only have a few minor comments with respect to grammar.

Page 12, line 104; add parentheses round abbreviation: (DOC)

Respond: We added parentheses as suggested.

Page 14, line 165-166; check sentence… parametric tests were used….. a nonparamtetric test was used….

Respond: The sentence was changed to: "The differences between means were examined using Tukey's HSD or Dunn tests for parametric or non-parametric analyses, respectively." (lines 164-165).

Page 20, line 324; check sentence…the decrease observed in the 1550 to 1700….

[revised manuscript text omitted]

[a] BET-$N_2$ Specific surface area

[b] Cation exchange capacity

[c] Organic carbon

[d] Obtained in soil aqueous extract (1:10 with 8.2 MΩ, Barnstead water)

[e] Humification index

[f] Fluorescence index

[Figure]

**Figure 1: The organic carbon (top), nitrogen (middle) and C:N (bottom), for equilibrated solutions (left) and solid phases after four reaction steps (right). Values for equilibrated solution OC and N represent cumulative removal from solution per soil mass. Dashed lines in OC and N plots show continuous-wet treatments, dotted lines in the C:N plot represent values of unreacted DOM solutions, error bars are the standard deviation, and letters indicate significant difference (ANOVA and Tukey's HSD $p$ <0.05) from unreacted control.**

[Figure]

**Figure 2. The fluorescence Index (FI), humification index (HIX) and specific UV absorbance at 245 nm (SUVA$_{254}$), for equilibrated**
**solutions reacted with JRB and SCM soils. The solid lines are wet-dry series, dashed lines are continuous-wet, and dotted lines are**
**unreacted DOM; error bars are the standard deviation.**

[Figure]

Continually Wet

Wet-dry

**Figure 3. Transmission FTIR spectra of the DOM dried solution reacted with JRB soils from steps 1 to 4 for continuous-wet (left) and wet-dry cycled (right) and the unreacted JRB DOM solution (bottom black line).  For color rendering of this image please refer to the online version.**

[Figure]

[Figure]

**Figure 4. Transmission FTIR spectra of the DOM dried solution reacted with SCM soils from steps 1 to 4 for continuous-wet (left)**
**and wet-dry cyclied (right) and the unreacted SCM DOM solution (bottom line). For color rendering of this image please refer to**
**the online version.**

[Figure]

**Figure 5. JRB soil reacted with DOM under wet-dry cycling. Left, C NEXAFS spectra extracted from C, Ca, and Fe regions of**
**STXM map. Spectra of unreacted soil (top) and DOM solution (bottom) are presented. Dashed vertical lines point out C species.**
**Right, tri-colored STXM map of fine fraction from JRB soil reacted four times with DOM under wet-dry cycling; Fe (red), Ca (blue)**
**and C (green). Image size 25 x 25 μm. For color rendering of this image please refer to the online version.**

[Figure]

Figure 6. JRB soil reacted with DOM under continuous-wet conditions. Left, C NEXAFS spectra extracted from C, Ca, and Fe regions of STXM map. Spectra of unreacted soil (top) and DOM solution (bottom) are presented. Dashed vertical lines point out C species. Right, tri-colored STXM map of fine fraction from JRB soil reacted four times with DOM during the continuous-wet treatment. Fe (red), Ca (blue) and C (green). Image size 25 x 25 μm. For color rendering of this image please refer to the online version.

[Figure]

**Figure 7. C NEXAFS extracted from C (red in Fig 6) regions of STXM map for the second step of the continuous-wet treatment (left) and from all 4 steps of the wet-dry treatment (right). For color rendering of this image please refer to the online version.**